# Les Houches guide to reusable ML models in LHC analyses

*Jack Y. Araz*[1]*, Andy Buckley*[2]*, Gregor Kasieczka*[3]*, Jan Kieseler*[4]*, Sabine Kraml*[5]*, Anders Kvellestad*[6]*,*
*Andre Lessa*[7]*, Tomasz Procter*[2]*, Are Raklev*[6]*, Humberto Reyes-Gonzalez*[8,9,10]*, Krzysztof Rolbiecki*[11]*,*
*Sezen Sekmen*[12]*, Gokhan Unel*[13]

[1] Jefferson Lab, Newport News, VA 23606, USA

[2] University of Glasgow, Glasgow, UK

[3] Univ. Hamburg, Germany

[4] Karlsruhe Institute for Technology, Karlsruhe, Germany

[5] Univ. Grenoble Alpes, CNRS, Grenoble INP, LPSC-IN2P3, 38000 Grenoble, France

[6] University of Oslo, 0316 Oslo, Norway

[7] Universidade Federal do ABC, Santo André, 09210-580 SP, Brazil

[8] Department of Physics, University of Genova, Via Dodecaneso 33, 16146 Genova, Italy

[9] INFN, Sezione di Genova, Via Dodecaneso 33, I-16146 Genova, Italy

[10] Institut für Theoretische Teilchenphysik und Kosmologie, RWTH Aachen, 52074 Aachen, Germany

[11] Faculty of Physics, University of Warsaw, 02-093 Warsaw, Poland

[12] Department of Physics, Kyungpook National University, Daegu, South Korea

[13] U.C. Irvine, Physics & Astronomy Dept., Irvine, CA, USA

### Abstract

With the increasing usage of machine-learning in high-energy physics analyses, the publication of the trained models in a reusable form has become a crucial question for analysis preservation and reuse. The complexity of these models creates practical issues for both reporting them accurately and for ensuring the stability of their behaviours in different environments and over extended timescales. In this note we discuss the current state of affairs, highlighting specific practical issues and focusing on the most promising technical and strategic approaches to ensure trustworthy analysis-preservation. This material originated from discussions in the LHC Reinterpretation Forum and the 2023 PhysTeV workshop at Les Houches.

### Keywords

BSM; Tools; Machine-learning; Reinterpretation.

## 1  Introduction

While the preservation and reuse of traditional "cut-and-count" collider analyses is increasingly stand-ard [1, 2], the preservation and reuse of analyses that employ machine-learning (ML) techniques is still in its relative infancy. Despite the growing reliance of experiments on ML-based analyses, there still remain serious obstacles to their practical reuse and long-term stability.

Machine-learning models are used in many different contexts within an experiment [3–7]. Some of these uses, such as $b$-tagging or electron/photon identification, are attempting to reconstruct a well-defined property of the physical event and can be relatively well-modelled using efficiency functions. In other cases, however, the ML model defines the analysis output, for example a signal-vs-background score, so that reinterpretability of the analysis directly depends on the reusability of the ML model.[1] There also exist cases in between (e.g. an analysis-specific top-tagger trained specifically on signal samples, where the efficiencies may differ significantly for different signal models), and all these use cases can take a wide variety of different inputs requiring different amounts of detector simulation/emulation.

Accessible and reusable ML information is thus a necessity that is increasingly recognised in the HEP community. For example, the Snowmass 2021 white paper on *Data and Analysis Preservation, Recasting, and Reinterpretation* [8] explicitly recommends:

> "…that reinterpretability and reuse should be kept in mind early on in the analysis design. This concerns, for instance, the choice of input parameters in ML models, the full specific-ation of the fiducial phase space of measurement in terms of the final state, including any vetos applied, and generally the choice of non-overlapping regions and standard naming of shared nuisances to facilitate the combination of analyses."

A few LHC analyses have recently published ML models on HEPData [9, 10]. This is a major step in the right direction. In this paper, we review the present status and discuss what is needed to make ML models, as often employed in LHC analyses, usable in reinterpretation studies. Our aim is to provide concrete guidelines for appropriate ML design, the serialisation and communication of the learned model, and the provision of validation material needed for the correct usage of reinterpretation tools. This concerns measurements and searches alike.

Following the guidelines of this paper will benefit not only the phenomenology community but also the analysis preservation efforts within the experimental collaborations by extending the usefulness of the experimental work. Furthermore, it can benefit the education of the next generation of physicists.

## 2  Mechanisms and examples

Standards exist for a degree of ML interoperability. For neural networks (NNs), the most common format is Open Neural Network Exchange (ONNX) [11], though, within ATLAS in particular, the Light-Weight

---

[1]Throughout this document by reinterpretability we mean reusability of the ML model or the analysis to infer results for scenarios not originally considered by the experimental analysis.

Neural Network (LWTNN) [12] format is also used. For boosted decision trees (BDTs), possible output formats include dependency-free serialisation through petrify-bdt [13] and treelite [14], custom formats like XML files from ROOT TMVA[^2] [15, 16], and to some extent also ONNX. Which output formats are available is likely to depend on the training environment, meaning that consideration to the preservation strategy is needed from an early stage of the ML-analysis design. There also exist ML architectures for which there is no current framework-independent representation, such as elements of generative networks used in normalizing-flow methods: this challenge is hence one that can be expected to continue, and the practicalities of preservation and reuse need to be considered when choosing an ML architecture.

Successful publication of an ML model requires not only sharing the model itself (including architectures, weights, and complete specification of any software dependencies) but also the detailed specification of the input data and any preprocessing performed (which to an extent can be encoded in ONNX), along with material to validate the model. Moreover, the long-term stability of the preservation format needs to be addressed – i.e. that model files saved using, for example, a version of ONNX from 2023 will still produce identical results in 2050. For this reason, it is useful to advertise the exact software versions used to produce, save and run the neural network. However, even with exact versioning, the long-term future ability to run such legacy code is not guaranteed, in particularly within frameworks that may or may not be maintained in the long term. Nonetheless, detailed documentation of the model used as well as validation material can greatly increase the future re-usability of analyses. A few initial encouraging steps are already being taken by the experimental collaborations in this direction.

Below we give some examples of analyses that have published ML information. The list is dominated by ATLAS analyses due to the current volumes of public information, but CMS is expected to follow suit soon.

- The ATLAS search for squarks and gluinos in final states with jets and missing transverse momentum, ATLAS-SUSY-2018-22 [17], has published BDT weights in XML format for use in ROOT TMVA on HEPData [18]. This is accompanied by a code snippet [19] with the implementation of the analysis selection at the truth level (also included in the SimpleAnalysis framework [20, 21]). Observed BDT score distributions in control and signal regions are provided for validation. This is an example of good practice which can readily be reused.
- The ATLAS search for $R$-parity-violating supersymmetry, ATLAS-SUSY-2019-04 [22] published trained networks in ONNX format on HEPData [23]. This analysis uses 64 high-level variables plus a continuous $b$-tag score from a ML-based tagger. Also in this case, a truth-level implementation is available [24], which proved essential for understanding the input variables. However, it is unclear how to reproduce the $b$-tag score, and how to verify that physics objects from any given fast-simulation package produce the intended ML responses within acceptable uncertainties.
- Another analysis for which the trained NN has been published in the ONNX format is the ATLAS search for supersymmetry in final states with many $b$-jets and missing transverse energy, ATLAS-SUSY-2018-30 [25] (targeting gluino decays to 3rd generation quarks and electroweakinos). Here, the ONNX files are contained in the SimpleAnalysis distribution [21]. The SimpleAnalysis code also details input variables and their normalisation. For $b$-tagging, a binary variable is used. The NN returns scores for evaluation of signal and control regions, however the latter are not included in SimpleAnalysis code and therefore cannot be used in recasting tools, e.g. for validation or for reuse with alternative signal models. Publishing also distributions of NN score for benchmark signal events would assist validation.
- The ATLAS search for neutral long-lived particles that decay into displaced hadronic jets in the calorimeter ("CalRatio LLP search") ATLAS-EXOT-2019-23 [26] published ONNX records of the NNs [27] as well as pure-C++ and pure-Python standalone executables of the BDTs [28] (preserved with petrify-bdt) on HEPData. These employ a large amount of low-level detector information

[^2]: Toolkit for Multivariate Data Analysis with ROOT.

and are not intended to be used outside the collaboration; the publication on HEPData is purely for the purpose of preservation. However, for public use, the analysis also provides 6-dimensional efficiency maps [29] parametrising the BDT+NN selection (plus example code to read and use them [30]). We will return to this kind of "model surrogates" in Section 5.

- The ATLAS anomaly-detection search in hadronic final states ATLAS-HDBS-2019-23 [31], which targets new resonances decaying into a Higgs boson and a generic new particle $X$, published the post-training weights from their variational recurrent NN (VRNN) as a PyTorch .pth file [32] on HEPData. Also provided is the VRNN Python code [33], but no description of the input/output variables, nor any description of how to use the code.

We also note that three of the above analyses have been presented in RAMP ('Reinterpretation: Auxiliary Material Presentation') seminars: slides and recordings are available online [34–36].

On the reinterpretation tools side, all the major frameworks (CheckMATE [37], GAMBIT's Collider-Bit [38], MadAnalysis 5 [39], Rivet [40], and ADL/CutLang [41, 42]) have developed interfaces for using published ML models. This was extensively discussed at the last two workshops of the Reinterpretation Forum: in December 2022 at CERN [43] and in August 2023 at Durham University [44].

Three of the above-mentioned ML-based searches have been implemented in recasting tools, with various degree of success. For models published in the ONNX format, the ONNX interface has been realised in all cases using the C++ ONNXRuntime library [45]. In particular, the ATLAS multi-$b$ jet-plus-MET analysis, ATLAS-SUSY-2018-30 [25], has been implemented in CheckMATE, Rivet, GAMBIT and ADL. The validation in Rivet, GAMBIT and CheckMATE showed generally good agreement with the cut-flows provided by ATLAS, though CheckMATE found up to 30% lower efficiencies than reported by ATLAS (the discrepancy is due to $b$-tagging efficiency); and the ADL/CutLang validation is ongoing.

These same four tools, as well as MadAnalysis 5, have also implemented ATLAS-SUSY-2019-04 [22]. All these implementations, however, have noted apparent tensions with the published cut-flows even before applying the NN score, with similar issues encountered using ATLAS' own public SimpleAnalysis code. The NN score distribution for a sample of signal events revealed significant discrepancies to the corresponding ATLAS distribution, and high sensitivity to the truth-level schemes used for emulation of the $b$-tag score input feature. More information is definitely needed for a reliable implementation of this analysis. Finally, CheckMATE has also implemented ATLAS-SUSY-2018-22 [17] using ROOT TMVA for BDT classification, and found good agreement with the ATLAS cut-flows.

These concrete implementations in public reinterpretation frameworks all relied heavily on the ATLAS SimpleAnalysis codes available from HEPData, effectively as pseudo-code documentation as the descriptions in the respective paper texts were usually insufficient on their own. Although it does not seem necessary to provide such a detailed description in publications, a short text document detailing shape of input tensors, variables, output tensors (e.g. the NN output in ATLAS-SUSY-2018-30 is non-trivial), etc. would be a further great assistance. Integration of the corresponding naming conventions of inputs and outputs in the ONNX file may also be useful. These hands-on experiences with the currently available material also inform the guidelines elaborated below.

## 3 Analysis design

In the design of an ML-based analysis, a number of choices have to be made, which will significantly influence the ease of publication and subsequent reuse. It is thus worthwhile that reinterpretability and reuse be kept in mind early on in the analysis design.

**Choice of framework:** The first choice is likely to be which ML package to implement the network or tree in. It is essential that this package be open source, e.g. Tensorflow, Keras, PyTorch, scikit-learn, JAX, or even ROOT TMVA, or that, at the minimum, the selected package can save the ML model for execution in an open-source or framework-independent form.

Proprietary packages, such as NeuroBayes [46] – used in several public analyses e.g. Refs. [47,48] – or MATLAB-based packages are effectively impossible to reuse in any reinterpretation tool other than collaboration-internal containerized preservations such as RECAST [49], which are not generally available to the phenomenology community.

Changes to formats or default settings, even in minor framework-version updates, can fundamentally change ML model behaviours, so the more complex an execution framework and its dependency tree, the larger the risk of algorithm behaviour becoming dependent on framework versioning.

**Choice of preservation format:** The typical complexity of machine-learning models, particularly neural-network architectures, makes the choice of ML preservation format an important question. This is largely independent of the programming language from which the ML-training framework is used, but it is crucial that the chosen format be readable and executable from compiled languages, particularly C++. The majority of event-loop reinterpretation tools are implemented in C++, and these must be able to execute ML models directly rather than e.g. inefficiently embedding a Python interpreter. Other non-C++ language options may also arise and become dominant in future, e.g. the Julia language is seeing gradual uptake in use for certain ML tasks.

This frames the central importance of flexible data formats. Although the overwhelming majority of ML training takes place in Python, the Python "pickle" format is not suitable as a long-term storage format because of a) its specificity to Python, b) versioning sensitivities within pickle itself, and c) its embedding of dependencies on (specific versions of) dependency packages. Fortunately, it has become common for ML frameworks to support export to compiled form for production deployments, and this has motivated industry development of interchange formats which are also natural candidates for preservation use. We note, however, that the industry focus is on short-timescale shifts between platforms rather than long-term preservation. Most major ML packages from outside HEP, e.g. TensorFlow and PyTorch, now support preservation and reloading via the ONNX format, including in compiled frameworks.

LWTNN is a lightweight C++ neural network framework that originated in the HEP community and runs the familiar risk of dependence on a few time-poor individuals. While it is currently supported, is used in ATLAS for efficient deployment of several neural-net applications, and has the benefit of having a far smaller code dependency than ONNXRuntime, the absence of long-term commitment is a concern for preservation. This is an issue that could benefit from discussion within and between the experimental collaborations. Conversion to LWTNN's own JSON from ONNX is possible, while the inverse conversion is currently not. **ONNX is hence the current format with most institutional/industrial support and the strongest case for use as the primary preservation format for HEP neural nets.**

We note, however, that newer techniques such as normalizing flows (NFs) are gaining significant uptake for publishing likelihoods [50] and many other applications [51]. However, currently, NFs are not supported by ONNX, and active involvement from the HEP community might be necessary to achieve this. This reflects the constantly changing nature of this technology and its format ecosystem. This will likely evolve further, adding a requirement that HEP archival systems perform periodic re-exporting, versioning, and regression testing of analysis ONNX data files to maintain usability with the evolving tool set. (As with many issues in ML reinterpretation, this concern also applies to other archive formats such as HistFactory [52], HS3 [3] [53], or ROOT, but the relative complexity and novelty of ML methods make the issue particularly acute.)

We also note that ROOT TMVA has recently developed an experimental system, SOFIE [54], for converting ONNX-based network preservations to C++ for use with TMVA. This has yet to be evaluated for reinterpretation purposes: a further decoupling from the large dependency on ROOT itself would be of great interest to public reinterpretation frameworks, most of which are not based

---

[3]High Energy Physics Statistics Serialization Standard.

on ROOT.

While neural nets have been the main focus of recent reinterpretation issues, it is important also to note the usage of other methods, most significantly BDTs, which have, in fact, been used more than NNs at the analysis level. The HEP-oriented petrify-bdt package can convert BDTs from TMVA and sci-kit-learn into C++ or Python code, using only conditional branching and no dependencies to make a very stable read/execute-only record of a trained BDT.[4] In addition, the treelite library and format purport to be a universal exchange mechanism for BDTs, similar to the role played by ONNX for NNs; while this, like ONNX, does add a library dependency and potential long-term stability issues, it has support for the XGBoost and LightGBM BDT libraries (but not currently TMVA).

For both NNs and BDTs and beyond, preserved ML files should be tracked and stored as a full part of the analysis to ensure they cannot accidentally be lost or overwritten. In case files are larger than can be supported by HEPData, **the CERN-based Zenodo data archive is the natural platform for hosting and preservation of trained ML models** – stable links should be provided from the HEPData archive to Zenodo or similar records via DOI codes.

**Choice of network architecture:** After choosing the package, thoughts turn to network architecture and choice of inputs. Some very advanced architectures may not yet be convertible to ONNX format (and LWTNN is even more restricted), so this choice can impact preservability. Similarly, heavily customised layers or activation functions (e.g. custom "lambda" layers in TensorFlow) may not be well preserved. Testing whether porting the network to a format such as ONNX is possible at an early stage and should be included in continuous-integration (CI) testing of analysis-code development. This will minimise the amount of work needed later to ensure the network is preservable.

The size of ML models may also become a potential stumbling block. In global fits, typically, tens if not hundreds of analyses are used at the same time. The combination of many very large models, in terms of disk or memory footprint, may make them unsuitable for reinterpretation. If designing lighter models is not possible, techniques such as neural-network pruning could be considered.

**Choice of input features:** The choice of input features leads to a parting of the ways between how best to reuse networks. In the extreme case, analyses may design taggers that are based largely or entirely on detector-level data that reinterpretation tools cannot hope to reproduce accurately – reinterpretation of this class of ML tool is discussed in Section 5.

However, there also exist many taggers that mainly or entirely use reco-level data that can be reproduced fairly well by the simple forms of detector simulation/emulation used by reinterpretation tools (either Delphes [55] or some variant of 4-vector smearing as in Rivet [56], MadAnalysis SFS [57], or GAMBIT ColliderBit [38]). Examples of such event features include the kinematics of reconstruction-level jets or leptons and missing energy, which admit fairly simple response-function parametrisations and already form the basis of most "cut-and-count" analyses' reinterpretation.

In these cases, reinterpretation would be greatly aided by avoiding inclusion of just one or two low-level variables in an otherwise reproducible ML model. As discussed above, several current analyses include among their input features a continuous ML-score variable produced by a complex and closed-source $b$-tagging algorithm operating on detector-level quantities such as hits in layers of silicon tracker. Using such unreproducible input variables renders the entirety of the preserved analysis-level ML discriminant unreproducible, even if those features only play a subleading role in its behaviour.

This problem can be ameliorated by a small tweak to the analysis design: although this score

---

[4]This simpler representation has been seen to also greatly improve the speed and memory footprint with respect to the original.

(and other such variables) is not easy to emulate in reinterpretation frameworks, replacing the continuous score with a tagged/not-tagged boolean value for ML purposes can ensure the network input can be well-approximated using efficiency-based techniques.

**ML model design checklist:**
– Use machine-learning software that can be easily converted to a stable interchange format supported by open-source tools. The ONNX and LWTNN JSON formats are the current most stable options for neural networks.
– Alternatively, if possible, export the ML model to executable code without dependencies beyond standard libraries.
– Preserved networks should be runnable with as few dependencies as possible from an API to a compiled language (e.g. C++), not just from Python.
– Avoid over-complexity in network design, e.g. not using customised layers or custom activation functions if the application does not require them. Ensure the chosen architecture has sufficient preservation-format support, particularly with ONNX.
– Where possible, and especially if the model is dominated by simple kinematic inputs, avoid input features that are heavily dependent on detector and reconstruction details.
– Where inputs are heavily detector-based, in addition to preserving the ML model itself, provide detailed efficiency maps (including mistag rates) or an equivalent surrogate network using less detector-sensitive input features (see Section 5).

## 4 Material for implementation and validation

Input features for ML algorithms should be described with sufficient detail to unambiguously recreate them, just like any other variable that is used directly to define a cut in an analysis. In addition, the exact format of the input features must be provided: an ML algorithm whose inputs are not well-defined is not usable! Specific issues include the parameter ordering (if the input to the network is a vector) or their exact keys (if it is a map), along with the units, conventions used (for example for angles), and any normalisation or padding information.

Possible formats to convey this information include auxiliary tables, an accompanying documentation note, README files, etc. Code snippets, like SimpleAnalysis implementations, are particularly helpful, as they automatically provide the needed information. A welcome example was the SimpleAnalysis routines provided in Ref. [22, 24] – we strongly recommend that such routines are saved to HEPData.

For validation, a plain data file of representative feature vectors and their corresponding ML output values would serve as a purely technical validation dataset (nb. this will also help ensuring version stability). Moreover, detailed cut-flow tables or histograms for the relevant signal model(s), including immediately before and after any ML-based cuts, can help validate both the network inputs and outputs. Plots of the model's output(s) assist in checking that the network has been implemented correctly, as the pass/fail in a cut flow may not be sufficiently informative. For example, as mentioned above, the BDT score distributions provided by ATLAS-SUSY-2018-22 proved very helpful. Similarly, plots of kinematic distributions *after* ML-based cuts can also be important, especially if the analysis goes on to make further kinematic cuts. As in other recasting situations, explicit configuration details for MC event-generator configurations allow direct comparison with plots in the paper or supporting documentation.

Some understanding of feature importance would also be very helpful, especially if accompanied by plots illustrating the distributions of the most important inputs and the NN output. This will allow the reinterpretation effort to focus on the most central variables for increased fidelity. As an example, see the discussion in Ref. [58], even though this network has not been made public. However, for this type of plot, it is very important to be very explicit about the validation sample being used in the plot – is it

from the samples used in training (and if so, how do we re-generate them), or is it representative of the events reaching that stage in the analysis?

While this requested information is extensive, we note that it is not a great extension of the information already regarded as most important for "traditional" cut-and-count analysis validation and that the extras are motivated by the "black box" inscrutability of ML observables offering relatively few physics sanity-checks, as well as the increased risk of corruption by framework translation or evolution. The information is available internally, but not always in a convenient form: we urge experiments to consider the pipeline by which it can be efficiently made available in a way usable outside the experiment code-base.

Finally, we emphasise that for much validation information, specific formats and polished presentation are not essential: it is much more important that it be available at all, than to erect barriers to publication which in practise only ensure that they remain publicly unavailable.

**Documentation/validation checklist:**

- Provide exact definitions (including units, ordering and conventions) of network input features, either as code examples or documentation.
- Provide a sample of input features and output values for technical validation.
- Provide plots of input and output variables for validation samples, if possible, with some indication of feature importance, as analysis supporting data.
- Provide cut-flow information, both before and after any ML-based selections.
- Validated and runnable published analysis code can be the clearest expression of both the general analysis logic and the specific interfacing with the ML functions.
- Full descriptions of the physics models used to generate the information above, e.g. SLHA files and generator run cards, are essential inputs to validating any serious reinterpretation.

## 5  Surrogate models

In some cases, the inputs to a trained ML algorithm will not be realistically accessible to those outside the experiment – examples include detector-level inputs, such as hits, or variables which are heavily dependent on detector geometry, such as soft jet-substructure variables. Examples of this sort of network include the $b$-jet taggers in both ATLAS and CMS [59, 60].

To first order, this can be approximated using truth-level efficiencies – this approach has been used by reinterpretation tools for both $b$-tagging and tagging for SM resonances. If using this approach, providing information for type-II errors, i.e. mistag rates, such as light or charm jets in $b$-tagging, is very important. Similarly, if the efficiency has any strong kinematic dependence, providing the efficiency in a set of kinematic bins – for example, $p_T$ and $\eta$ – would be necessary. An outstanding example are the 6D(!) efficiency maps parametrising the BDT+NN selection of the ATLAS CalRatio LLP search [26] mentioned in section 2.

It would be worthwhile to explore if the accuracy of modelling could be improved using surrogate ML models, or surrogates – neural networks trained to replicate the output of the original tagger but using input events with a simpler set of attributes, typical as used in the reinterpretation frameworks. For all surrogates it would be crucial to provide a quantification of the surrogate's uncertainty.

What inputs the surrogate should take will depend on the inputs of the original network, i.e., the physics the network is trying to capture, and the level of accuracy required of the surrogate. Broadly speaking, we can split the different sets of inputs a surrogate may take into three classes depending on the level of information required as input:

- **Reco-level surrogates:**  features of reconstructed (clustered) objects after showering, hadronization and detector emulation, such as jets, dressed leptons and missing energy. In most cases

('simple reco-level') these features will include the object kinematics; which are available at some level of precision, from public detector-emulation and clustering tools like FastJet, Delphes or the Rivet/MadAnalysis 5 convolution-based "smearing" systems. Adding in more complex reco-level variables, such as jet substructure, is also possible, but may stretch the capabilities of fast detector emulation.

- **Particle-level surrogates:** 4-vectors of all particle-level objects after showering and hadronisation (with or without detector smearing), i.e. hadrons, leptons, photons or long-lived Beyond-the-Standard Model (BSM) particles. This level may also include information about the positions and timings of decay or production vertices.

- **Parton-level surrogates:** 4-vectors of parton-level particles produced by the MC generator prior to hadronisation or detector emulation. This level can be particularly useful when building surrogate models involving long-lived BSM particles. Whether or not showering needs to be included depends on the physics case and on whether initial-state radiation is included in the hard scattering process.

Which of these to use will depend heavily on the use case. For example, "reco-level" surrogates clearly present the easiest task to neural networks, but this may rely on detector emulation providing reliable results. At present, this is not always the case, e.g. for soft jet-substructure features. Methods that require extensive jet reclustering may also be unsuited to reinterpretation tools that prioritise performance.

A clear distinction should also be drawn between surrogates that provide the true label (e.g. a boolean variable expressing if a jet actually contains a $b$-hadron), and those that do not. The surrogates that do, effectively become very well-parameterised efficiency maps, and will likely perform better. However, there may be valid reasons not to provide this information to the surrogate: for example, if a tagger is being used to tag BSM particles directly, then directly checking the event record for a particular BSM particle-ID code becomes dangerously model-dependent and hence cannot be reinterpreted beyond the original model. Such implications again hint at questions to be asked at the analysis-design stage, about how to maximise the general-purpose usefulness and hence the long-term impact of a physics study.

The field of preserving analyses using surrogates is even younger than the other content of this note. In the future we hope to see more examples – among them several projects started at the 2023 Les Houches PhysTeV workshop – that can shape best practices for analyses going forward. That being said, analyses should not shy away from making their own surrogates immediately! Of course the guidelines for design, documentation and validation of ML models elaborated above also apply to surrogates.

**Surrogates checklist:**
- If the ML model in an analysis is not suitable for sharing and reuse in its original form, detailed efficiency map or a trained surrogate model are necessary to enable reinterpretation.
- For providing a surrogate model, follow the analysis design and documentation/validation guidelines above.

# 6  Summary and conclusions

Machine-learning methods have great potential to boost the scientific sensitivity of physics experiments, and are undoubtedly here to stay as a major part of how we do physics. But when applying ML-based methods we should bear in mind that the long-term impact of an analysis depends not only on its sensitivity to the initial physics model, but also on its future application to other models. This, combined with the large investments in major collider facilities and data-taking, demand the possibility for data reuse rather than publish-and-forget, and so priority should be given to the long-term and framework-independent re-usability of ML methods. In this note we have highlighted practical issues (and currently preferred partial solutions) to be actively considered as early as possible in ML-based analysis design

and implementation. We anticipate future updates of these recommendations as technological capabilities and physics challenges evolve, but an early adoption of the present recommendations will go a long way towards ensuring long-lasting scientific impact of current and near-future ML-based analyses.

## Acknowledgements

AB acknowledges support by the UK STFC Consolidated Grant scheme (ST/W000520/1) and TP an STFC doctoral training studentship. The work of JK is supported by the Alexander von Humboldt-Stiftung. The work of SK is supported in part by the IN2P3 master project "Théorie – DATAMatter"; financial support by the CNRS Formation Permanente for attending the Les Houches workshop is also gratefully acknowledged. AK and AR are supported by the Research Council of Norway (RCN) through the FRIPRO grant 323985 PLUMBIN'. AL is supported by FAPESP grants no. 2018/25225-9 and 2021/01089-1. HRG is supported by the Deutsche Forschungsgemeinschaft (DFG, German Research Foundation) under grant 396021762 – TRR 257: Particle Physics Phenomenology after the Higgs Discovery. HRG also acknowledges the support from the Italian PRIN grant 20172LNEEZ. KR is supported by the Norwegian Financial Mechanism 2014-2021, grant no. 2019/34/H/ST2/00707; and by the National Science Centre, Poland grant 2019/35/B/ST2/02008. The work of SS is supported by the Basic Science Research Program through the National Research Foundation of Korea (NRF) funded by the Ministry of Education under contract NRF-2021R1I1A3048138.

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
