# Peer review of "Les Houches guide to reusable ML models in LHC analyses"

_SciPost Physics Community Reports_

## Round 2 · Referee Report · Anonymous (Referee 1) · 2024-3-14

Strengths

1- Good overview of the subject
2- Provides possible strategies to cope with the issues of Machine Learning model preservation

Report

The document contains a report on the topic of reusable Machine Learning models in collider analyses. The material originated from the LHC Reinterpretation Forum and the 2023 PhysTeV workshop at Les Houches.

The main purpose of the report is to highlight the main issues related to the preservation of ML models used in LHC analyses and discuss possible strategies to cope with them. The relevance of the discussion is due to the fact that an increasing number of collider analyses involve ML techniques (in various forms), but no common preservation standard has been developed so far to ensure later reusability. Critical issues are related to the format of ML model storage together with the choice of validation data needed to ensure the reproducibility of the analyses. These aspects are discussed in section 2 together with some example from published analyses from the ATLAS collaboration.

A review of the frameworks for ML model design and storage is contained in section 3, with details on the currently available preservation formats, architecture design and choice of input data. The issue of validation protocols and validation data storage is addressed in section 4. Finally section 5 contains a discussion on surrogate models, which are needed for ML reusability in the case in which the original analysis uses input data that are not publicly available (eg. detector-level data). All these sections contain some recommendations and possible strategies to address the highlighted issues.

I think that the report provides an interesting review of the topic, giving a good critical overview of the current state of the subject and of the main issues that should be addressed to ensure long-term preservation of the ML-based LHC analyses. The document could be potentially interesting for the experimental community and could stimulate efforts to design a robust and common standard for ML model preservation.

---

## Round 2 · Referee Report · Anonymous (Referee 2) · 2024-5-15

Strengths

The paper is good for documentation and is asking for some standardization of Machine Learning models. So, in principle, it is a “best practice” paper. However, the authors only give examples of what others have done and don’t really develop protocols of their own. It is more of a review of practices that others have implemented and the authors think they are good to adopt.

Weaknesses

There isn’t much innovation in this paper and it more or less reads like a documentation of some discussions the authors had. At the technical level, it’s a bit inaccessible to those who are not experts in the field by using too many acronyms that the authors do not spell out. The point is that experts will already know what the authors are recommending and the document gets complicated to read for others to whom it might be potentially useful.

Report

Since Machine Learning is seeing an increased uptake in mainstream particle physics analyses, a "best practices" paper is relevant to the community. However, sharing algorithms and models along with data used for training and standardized tests of the models is a practice that is well established in the machine learning and computer science/engineering communities. While this paper puts this in the context of particle physics its applicability is limited to specific use cases where the models might be reused in several analyses or for validating/reproducing published results. For the latter, it is often more productive to share the training data and the specifications of the model architectures. I can recommend this article for publication with minor edits that I list below.

Requested changes

- The authors use a lot of technical acronyms without defining them explicitly. This makes the document accessible only to those who are familiar with the technicalities and they are the readers who least need this document. For the readers who are not familiar with the acronyms a glossary or some sort of explanation of the the acronyms used would significantly increase the readability of the article given the broader audience of this journal.

- the authors put in interpretability (in terms of feature importance) as an afterthought although, in my opinion, it is quite important as a validation of an ML model. They do not refer to any of the vast body of work done in interpretable machine learning and feature importance in particle physics. For the only article they refer to, they are a bit vague about which plot they are discussing. Being more specific will help here and if the authors wish to discuss the interpretability of ML models, they should put in more than a couple of sentences and refer to the existing literature.

Recommendation

Ask for minor revision

  • validity: good
  • significance: good
  • originality: ok
  • clarity: ok
  • formatting: good
  • grammar: good

Author:  Tomasz Procter  on 2024-09-11  [id 4757]

(in reply to Report 2 on 2024-05-15)

Apologies for the delayed response. Following the comments, we have made some changes (resubmitted to arXiv, should be circulated tomorrow, September 12th). See below for a more detailed response to some of the referees comments:

  • REFEREE: The authors use a lot of technical acronyms without defining them explicitly. This makes the document accessible only to those who are familiar with the technicalities and they are the readers who least need this document. For the readers who are not familiar with the acronyms a glossary or some sort of explanation of the the acronyms used would significantly increase the readability of the article given the broader audience of this journal.

REPLY: We have now defined ONNX, LWTNN, TMVA, HS3 and BSM in the text or in footnotes. As for other technical `acronyms' (like ROOT, CheckMATE, Rivet, etc.), these are software packages of wide-spread use in our field, so we don't define them but provide the relevant references. We think that our target audience is sufficiently familiar with these software tools so that a broader discussion is not needed (and beyond the scope of this short paper); in any case, detailed explanations are given in Ref. 2, which we cite right at the beginning of the introduction.

  • REFEREE: the authors put in interpretability (in terms of feature importance) as an afterthought although, in my opinion, it is quite important as a validation of an ML model. They do not refer to any of the vast body of work done in interpretable machine learning and feature importance in particle physics. For the only article they refer to, they are a bit vague about which plot they are discussing. Being more specific will help here and if the authors wish to discuss the interpretability of ML models, they should put in more than a couple of sentences and refer to the existing literature.

REPLY: interpretable (and reproducible) machine learning is a highly interesting topic in itself, but not the topic of our paper. Here, we are primarily concerned with the reuse of ML models for physics studies. In order not to take attention away from the primary purpose of our paper, we do not wish to enter into any deeper discussion of the interpretability of ML models.

Other changes:
- References provided for HistFactory and HS3 (now Refs. 52 and 53); - References 2 and 44 polished

---

## Editorial Decision

editorial_decision: